# The Influence of Anandamide on the Anterior Pituitary Hormone Secretion in Ewes—Ex Vivo Study

**DOI:** 10.3390/ani10040706

**Published:** 2020-04-17

**Authors:** Dorota Tomaszewska-Zaremba, Karolina Wojtulewicz, Kamila Paczesna, Monika Tomczyk, Katarzyna Biernacka, Joanna Bochenek, Andrzej Przemysław Herman

**Affiliations:** The Kielanowski Institute of Animal Physiology and Nutrition, Polish Academy of Sciences, Instytucka 3, 05-110 Jabłonna, Poland; k.wojtulewicz@ifzz.pl (K.W.); kopycinskakamila@gmail.com (K.P.); m.tomczyk@ifzz.pl (M.T.); k.biernacka@ifzz.pl (K.B.); j.bochenek@ifzz.pl (J.B.); a.herman@ifzz.pl (A.P.H.)

**Keywords:** anandamide, gonadotropins, anterior pituitary, ewe

## Abstract

**Simple Summary:**

Mankind has been using *Cannabis sativa* (marijuana) for a century, but the system of endogenous cannabinoids has only been recently described. This system modulates an array of physiological and psychological functions, including reproduction. Interaction between gonadotropin-releasing hormone (GnRH) neurons and the endogenous cannabinoid system plays an important role in this process. The presence of cannabinoid receptors in the pituitary raises a presumption that anandamide, the endogenous cannabinoid, may influence reproduction process directly in the anterior pituitary. The aim of the undertaken studies was to investigate the effect of anandamide on gonadotropins secretions from anterior pituitary explants downloaded from ewes. It was demonstrated that anandamide inhibited gonadotrophic hormones secretion from the anterior pituitary explants. This could suggest that negative effects of cannabinoids on reproductive processes could be caused by the direct action of these compounds at the pituitary level.

**Abstract:**

Cannabinoids (CBs) are involved in the neuroendocrine control of reproductive processes by affecting GnRH and gonadotropins secretion. The presence of cannabinoid receptors (CBR) in the pituitary raises a presumption that anandamide (AEA), the endogenous cannabinoid, may act on gonadotrophic hormones secretion directly at the level of the anterior pituitary (AP). Thus, the aim of the study was to investigate the influence of AEA on gonadotropins secretions from AP explants taken from anestrous ewes. It was demonstrated that AEA inhibited GnRH stimulated luteinizing hormone (LH) and follicle stimulating hormone (FSH) secretion from the AP explants. Anandamide influences both *LH* and *FSH* gene expressions as well as their release. AEA also affected gonadoliberin receptor (GnRHR) synthesis and expression. The presence of CB1R transcript in AP explants were also demonstrated. It could be suggested that some known negative effects of cannabinoids on the hypothalamic-pituitary-gonadal axis activity may be caused by the direct action of these compounds at the pituitary level.

## 1. Introduction

The hypothalamic-pituitary-gonadal (HPG) axis controls reproduction process in females. Hypothalamus, where gonadotropin-releasing hormone (GnRH) is produced and then released to the hypothalamic-pituitary portal circulation, is the structure playing the most important role in the control of this neurohormonal system. GnRH is released in a pulsatile manner, which ensures normal, physiological secretion of luteinizing hormone (LH) and follicle-stimulating hormone (FSH) from the pituitary [1]. It is suggested that cannabinoids (CBs) influence the neuroendocrine control of reproduction, influencing the secretion of GnRH and gonadotropins, gonadal steroids production, spermatogenesis, ovulation, and sexual behavior [2,3,4,5]. Many previous studies on lower animals, as well as on humans, revealed that Δ9- tetrahydrocannabinol (THC), the main psychoactive component of marijuana, influenced pituitary hormone secretion. However, it is believed that the structure where inhibitory action of THC on pituitary hormones release (such as growth hormone (GH) and LH) takes place seems to be the hypothalamus [6,7].

The endocannabinoid (ECB) system is a known and well-characterized endogenous system in mammalian and non-mammalian vertebrates [8,9]. Endocannabinoids are synthesized and released in response to increased intracellular Ca^2+^ concentration [10]. Generally, ECBs are defined as the endogenous ligands of cannabinoid receptors, which could be esters and ethers of long-chain polyunsaturated fatty acid or amides occurring, among others, in the central nervous system (CNS), peripheral tissues, even in reproductive fluids [11]. The main ECBs are the N-arachidonoyl-ethanolamine (anandamide, AEA) and 2-arachidonoylglycerol (2-AG) [12]. Until now, there have been three cannabinoid receptors (CBRs) recognized: type 1 and type 2 receptors (CB1R and CB2R - encoded by *CNR1* and *CNR2* genes), and classical seven transmembrane G-protein coupled receptors, expressed in the CNS, the immune system, and several tissues including gonads [13,14,15,16]. The GPR55—orphan G coupled receptor, is now known as the other CB receptor [17]. Moreover, AEA could act as an intracellular ligand of the potential type 1 vanilloid receptor (TRPV1) channel [5].

It has been shown that ECBs, as well as exogenous cannabinoids, mainly THC, markedly modify the anterior pituitary (AP) hormones secretion [6]. Although this action of ECBs, including AEA, may be mediated mainly through an activation of CBRs in the hypothalamus, the presence of cannabinoid receptors in the pituitary [18] raises a presumption that AEA may also act on gonadotrophic hormones secretion directly at the level of this gland. The expression of CB receptors, as well cannabinoids biosynthetic, and hydrolyzing enzymes, have been found both in the pituitary *pars distalis* and in the pituitary cell cultures [3,19,20,21,22]. Moreover, in the studies on amphibians and mammals, it was observed that CBRs are found on the gonadotrophs, suggesting that cannabinoids can directly affect secretory activity of these cells [22,23,24]. It is worth mentioning that some measurable amounts of ECBs were found in the AP of male rats, which indicates that these compounds may be important for the autocrine and paracrine regulation of AP cells activity [19].

In the present study, the hypothesis that endocannabinoid, AEA, may influence the reproductive process in ewe at the level of the AP, affecting gonadotropins secretion, has been investigated. Therefore, the effect of AEA on LH and FSH secretions from AP explants, collected from anestrous ewes was determined. The expressions of *LH-β*, *FSH-β*, and gonadoliberin receptor (*GnRHR*) genes was also assayed in the AP explants.

## 2. Materials and Methods

### 2.1. Animals

The ex vivo experiment was carried out on the APs collected from adult, 3-year old Blackhead ewes (*n* = 6) in anestrous season. After euthanasia, the ovine brains were rapidly removed from the skulls, and the AP was dissected and divided into four fragments (explants). All procedures were accepted by the Local Ethics Committee of Warsaw University of Life Sciences—SGGW (Warsaw, Poland; authorization no. 18/2014) in confirmation to all the methods and guidelines of Act of 15 January, 2015 on the protection of animals used for scientific or educational purposes (Poland), following the international parameters on use and handling of animals.

### 2.2. Experimental Procedure

#### Incubation of Pituitary Explants

The explants incubation had been performed based on the methodology described in the previous study [25]. The explants were first pre-incubated for 1 h in 24-well plates (Becton Dickinson Labware, Franklin Lakes, NJ, USA) with medium 199 (Sigma-Aldrich, St. Louis, MO, USA). The medium was replaced every 15 min. The pre-incubation was performed to wash out the blood and hormones from the pituitary fragments. Next, all explants were incubated for additional 30 min in M199. Finally, the explants from each ewe were treated with: 1/M199 only (control explants), 2/GnRH (100 pmol/mL; Sigma-Aldrich, St. Louis, MO, USA) (a positive control), 3/AEA (100 µM; Sigma-Aldrich, St. Louis, MO, USA. The dose was chosen on the basis of preliminary study with doses of AEA 10^−6^, 10^−5^, and 10^−4^ µM where the dose of 10^−4^ µM was found optimal), 4/GnRH + AEA and incubated for 3 h at 37 °C (87% O_2_, 5% CO_2_). The incubation medium consisted of M199 HEPES with Earle’s salts, sodium bicarbonate, and HEPES (25 mM) with penicillin-streptomycin (10 mL/L) (Sigma-Aldrich, St. Louis, MO, USA). After 3 h of incubation, the explants were weighed and all explants and media were frozen at −80 °C until further assays.

### 2.3. Assays

#### 2.3.1. Radioimmunoassay for LH

The LH concentration in the medium was assayed with a double-antibody radioimmunoassay (RIA) using anti-ovine-LH (antibodies and standards provided by the National Pituitary Agency and Dr. A.F. Parlow; Harbor-UCLA Research and Education Institute c/o Los Angeles Biomedical Research Institute, Los Angeles, CA, USA, 90060) and anti-rabbit-ɤ-globulin antisera (Sigma-Aldrich, St. Louis, MO, USA) according to Stupnicki and Madej [26]. The assay sensitivity was 0.3 ng/mL and the intra- and inter-assay coefficients of variation were 8% and 11.5%, respectively.

#### 2.3.2. Radioimmunoassay for FSH

The concentration of FSH in the medium was determined by double antibody radioimmunoassay (RIA) using anti ovine-FSH (antibodies and standards provided by the National Pituitary Agency and Dr. A.F. Parlow; Harbor-UCLA Research and Education Institute c/o Los Angeles Biomedical Research Institute, Los Angeles, CA, USA, 90060) and anti-rabbit-ɤ-globulin antisera, according to L’Hermite et al. [27]. The assay sensitivity was 1.5 ng/mL and the intra- and inter-assay coefficients of variation were 3.5% and 11.3%, respectively.

#### 2.3.3. Relative Gene Expression Assay

The relative gene expression was assayed with the method earlier described by our team [1]. Shortly, a NucleoSpin^®^ RNA kit (MACHEREY-NAGEL GmbH and Co, Düren, Germany) was used to isolate the total RNA from the AP explants. All isolation steps were conducted in accordance with the manufacturer’s instruction. The purity and concentration of the isolated RNA was quantified spectrophotometrically with the use of a NanoDrop 1000 instrument (Thermo Fisher Scientific Inc., Waltham, MA, USA). The integrity of isolated RNA was confirmed by electrophoresis with the use of 1% agarose gel stained with ethidium bromide. The Maxima™ First Strand cDNA Synthesis Kit for RT-qPCR (Thermo Fisher Scientific, Waltham, MA, USA) was used to perform cDNA synthesis. As a starting material for cDNA reversed transcription reaction (RT), synthesis 2 µg of total RNA was used. A Real-Time RT-PCR was carried out with the use of the HOT FIREPol EvaGreen^®^ qPCR Mix Plus (Solis BioDyne, Tartu, Estonia) and HPLC-grade oligonucleotide primers (Genomed, Warszawa, Poland). The primer sequences were designed using Primer 3 software (Table 1). One reaction mixture of total volume amounting 20 µL contained: 4 µL of PCR Master Mix (5×), 14 μL of RNase-free water, 1 µL of primers (0.5 µL each primer, working concentration 0.25 µM), and 1 μL of the cDNA template. The reactions were conducted on a Rotor-Gene 6000 instrument (Qiagen, Dusseldorf, Germany) with the following protocol: 95 °C for 15 min to activate Hot Star DNA polymerase, 30 cycles of 94 °C for 5 s for denaturation, 56 °C for 20 s for annealing, and 72 °C for 15 s for extension. The specificity of the amplification was confirmed by a final melting curve analysis. The relative gene expression was calculated using the comparative quantification option [28] of the Rotor Gene 6000 software 1.7. (Qiagen, Dusseldorf, Germany). Three housekeeping genes were examined: glyceraldehyde-3-phosphate dehydrogenase (GAPDH), β-actin (ACTB), and cyclophilin C (PPIC). The mean expression of these three housekeeping genes was used to normalize the expression of the analyzed genes. The results are presented in arbitrary units, as the ratio of the target gene expression to the mean expression of the housekeeping genes. The average relative quantity of gene expression in the control group of the AP explants was set to 1.0.

#### 2.3.4. Western Blot Assays for GnRHR Expression in the AP

Before electrophoresis, the protein concentrations of samples isolated previously from the AP using the NucleoSpin^®^ RNA/Protein Kit (MACHEREY-NAGEL Gmbh & Co., Düren, Germany) were quantified using a Protein Quantification Assay Kit (MACHEREY-NAGEL Gmbh & Co., Düren, Germany). The appropriate volume of molecular grade water (Sigma-Aldrich, St. Louis, MO, USA) was added to such a volume of sample, which contained 50 µg of total protein to bring the total sample volume to 20 µL. Next, 19 µL of Laemmli buffer (Sigma-Aldrich, St. Louis, MO, USA) and 1 µL of β–mercaptoethanol (Sigma-Aldrich, St. Louis, MO, USA) were added. Such mixtures were boiled for 3 min. Electrophoresis was then performed in the presence of molecular weight markers (Spectra Multicolor Broad Range Protein Ladder, Thermo Fisher Scientific Inc., Waltham, MA, USA). Denatured samples and molecular weight standards were loaded onto 4–12% polyacrylamide gels and subjected to electrophoresis in a Tris-glycine running buffer using the Protean II xi Cell (Bio-Rad Laboratories, Inc., Hercules, CA, USA), according to the manufacturer’s instructions. Next, proteins were transferred in Tris-glycine blotting buffer to polyvinylidene difluoride membranes (ImmobilonTM-P (0.45 µm), Merck KGaA, Darmstadt, Germany) using aTrans-Blot^®^ SD Semi-Dry Transfer Cell (Bio-Rad Laboratories, Inc., Hercules, CA, USA) for 30 min at 20 V. The membranes were blocked for 1 h at a room temperature in blocking buffer made up of Tris buffered saline at pH 7.5 with 0.05% Tween-20 (TBST) (Sigma-Aldrich, St. Louis, MO, USA) containing 3% bovine serum albumin fraction V (Sigma-Aldrich, St. Louis, MO, USA). Next, membranes were incubated overnight at 4 °C with the following primary antibodies: goat anti-GnRHR polyclonal antibody (cat no. sc-8682, Santa Cruz Biotechnology Inc., Dallas, USA), and mouse anti-ACTB monoclonal antibody (cat no. sc-47778, Santa Cruz Biotechnology Inc., Dallas, USA) dissolved in blocking buffer at dilutions of 1:500 and 1:1000, respectively. After washing three times, membranes were incubated with the following secondary Horseradish Peroxidase Protein (HRP) conjugated antibodies: donkey anti-goat Immunoglobulin (Ig)G-HRP (cat no. sc-2304, Santa Cruz Biotechnology Inc., Dallas, TX, USA) and goat anti-mouse IgG1 heavy chain (HRP) (cat no. ab97240, Abcam, Cambridge, UK) dissolved in blocking buffer at a dilution of 1:10,000. After washing three times, the membranes were visualized using chemiluminescence detection with a Clarity™ Western ECL Substrate (Bio-Rad, Hercules, CA, USA) by a ChemiDoc™ MP Imaging System (Bio-Rad, Hercules, USA). Densitometric analysis of the membranes was performed using the Image Lab™ program (Bio-Rad, Hercules, CA, USA).

### 2.4. Statistical Analysis

The results of AEA treatment on the concentrations of LH, FSH in all types of media and on gene expression were analyzed by two-way (AEA and GnRH) analysis of variance, ANOVA (STATISTICA; ver. 13.1 Dell Inc., Round Rock, TX, USA) and followed by a post-hoc analysis with the Fisher’s least significance test. The results were considered statistically significant at P at least *p* ≤ 0.05. All data are presented as the mean ± SEM.

## 3. Results

### 3.1. The Ex Vivo Influence of GnRH and Anandamide on the LH and FSH Release from the AP Explants

In the AP explants from anestrous ewes, the GnRH treatment significantly (Fisher’s test *p* ≤ 0.001 for LH and *p* ≤ 0.05 for FSH) stimulated LH (Figure 1A) and FSH (Figure 1B) release. It was also found that anandamide treatment significantly (Fisher’s test *p* ≤ 0.05) suppressed the stimulatory effect of GnRH treatment on LH (Figure 1A) and FSH (Figure 1B) release. No changes in LH release vs. control group were found in the explants incubated with anandamide alone, significant suppression (Fisher’s test *p* ≤ 0.05) of FSH release after anandamide treatment was observed in comparison to the control group.

### 3.2. The Ex Vivo Influence of GnRH and Anandamide on the LHβ and FSHβ Gene Expression in the AP Explants

The GnRH treatment stimulated (Fisher’s test *p* ≤ 0.05) both the *LHβ* (Figure 2A) and *FSHβ* (Figure 2B) gene expression in the AP explants. The inhibitory effect of AEA on mRNA level for LHβ as well as for FSHβ vs control group (Fisher’s test *p* ≤ 0.05) and GnRH treated group (Fisher’s test *p* ≤ 0.01) was observed. In the explants co-incubated with GnRH, AEA suppressed (Fisher’s test *p* ≤ 0.01) stimulatory effect of GnRH on the *LH-β* and *FSH-β* genes expression.

### 3.3. The Ex Vivo Effect of GnRH and Anandamide on the GnRHR Gene Expression and GnRHR Protein Expression in the AP Explants

It was found that treatment with GnRH did not affect GnRHR mRNA expression in AP explants. The inhibition (Fisher’s test *p* ≤ 0.05) of *GnRHR* gene expression in all groups of explants treated with AEA was also noted (Figure 3A). In AP explants, the GnRH treatment did not affect GnRHR expression. However, AEA alone, as well as together with GnRH, suppressed (Fisher’s test *p* ≤ 0.05) GnRHR expression in comparison to control and GnRH groups (Figure 3B).

### 3.4. The Ex Vivo Effect of GnRH and Anandamide on the Gene Expression of CNR1 in the AP Explants

The presence of a *CNR1* transcript in AP explants from anestrous ewes was demonstrated in the current study. It was found that neither GnRH nor AEA affect CB1 mRNA expression in AP explants (Figure 4)

## 4. Discussion

The present study is the first to describe the effect of AEA on the gonadotropins secretion from ovine anterior pituitary explants. In our *ex vivo* study, we have demonstrated that AEA in a dose of 100 µM significantly decreased induced by GnRH release of LH and FSH in anestrous ewes. This data is partially in accordance with the results of the previous study conducted by Wenger et al. [31] in which AEA decreased the level of LH in the dispersed rats AP cells. In this experiment the inhibitory influence of AEA on LH release was prevented by CBR1 selective cannabinoid receptor antagonist—SR 141716 (SR). On the other hand, in contrast to the current study, Wenger et al. [31] did not find any effect of AEA on FSH concentration in the AP cells. We have also shown that AEA not only influenced the release of LH and FSH from the AP explants but also decreased the expression of *LH-β* and *FSH-β* genes. The ability of AEA to interfere with the gonadotropins transcription may be among the reasons of decreased release of these hormones from the pituitary gland. The results of *ex vivo* study suggests that the effect of ECBs on the secretory activity of AP cells could be similar to the action of exogenous cannabinoids. As it was previously demonstrated, that exogenous as well endogenous cannabinoids modify the anterior pituitary hormone secretion. THC as well as AEA decreased prolactin and gonadotropin release [32] and increased ACTH plasma levels [33]. Wenger et al. [31] described direct effect on the anterior pituitary hormone secretion. They demonstrated that selected doses of AEA caused a decrease of LH and prolactin and increase of ACTH and GH releases by the AP cells dispersed in vitro. Cannabinoids have been known as negative regulators of reproduction in rodents, primates, as well as in humans. It was described that chronic administration of CBs to prepubertal female rats could decrease serum content of LH and sex steroids [34]. It was also shown in experiment on ovariectomized rats that the acute THC administration inhibited pulsatile LH secretion and blocked steroid-mediated positive feedback on the LH surge [35]. However, it should be mentioned that in some studies, the role of cannabinoids on the gonadotropins secretion is suggested to be more elusive. Scorticati et al. [36] observed that AEA injections in ovariectomized estrogen primed (OVX-E) rats caused an increase of plasma LH levels. This was probably due to increased GnRH release; it was observed that AEA evoked GnRH release from mediobasal hypothalamus (MBH) of OVX-E rats. The effects of cannabinoids on FSH secretion are more ambiguous. It was observed that the use of chronic marijuana reduced the level of circulating FSH in human males; however, no effects of THC or marijuana on FSH levels were also reported [37]. It is worth mentioning that the influence of exogenous and endogenous CBs on the gonadotropins secretion may partially result from the action of these compounds at the level of hypothalamus. It was shown that cannabinoids inhibit GnRH release [36] and GnRH transcription [11,38]. An ambiguous effect of AEA on both gonadotropins and GnRH secretion may be connected with the daily fluctuations of CBs receptor expression. In experiments on female rats it was demonstrated that in the anterior pituitary the *CNR1* gene expression changes during the estrous cycle—the highest values occurred during estrous while the lowest—on the first day of diestrous and in proestrous [20].

The results of our study also suggest that one of the mechanisms through which AEA could affect LH and FSH secretion from AP could be down-regulation of GnRHR. We observed decreased GnRHR mRNA and protein expression in the AP explants after AEA treatment. As was demonstrated previously, the reduction of GnRHR on pituitary cells accompanied reduced secretion of LH [39,40,41,42].

We have also observed that AEA influenced gonadotropins secretion acting via the CB1R receptor. A significant expression of CB1R mRNA in AP explants was also noted but we did not observed any effect of AEA on the CN1R gene expression. This indicates that CB1R may be involved in the effect of AEA on the anterior pituitary cells activity. This observation is generally consistent with the results of previous studies in which the CB1R receptor was demonstrated in rat pituitary [22]. The mRNA for CB1R receptor mRNA was observed in the anterior pituitary, mainly in the intermediate lobe, but no transcripts were shown in the neural lobe [19]. The existence of CB1R receptors in the anterior pituitary was also evidenced in the study of Dall’Aglio et al. [43] on rabbits. It is worth mentioning that all CB1R immunoreactivity was co-localized in cells showing strong nuclear immunoreaction for estradiol receptor [43]. Chakrabarth et al. [44], in the study on mice, showed that AEA increased the whole brain gene expression of the *CN1R* in the ICR strain of mice but not in the C57BL/6 animals. In turn, in the DBA/2 mice, AEA slightly reduced the *CNR1* gene expression in brain. In other studies on male rats, the acute and chronic administration of AEA analog- R-methanandamide (AM356) did not affect mRNA of CB1R receptor in studied hypothalamic structures [45]. Confirmed in our study, mRNA for the CB1R in the anterior pituitary explants in anestrous ewes supports the hypothesis assuming the effect of cannabinoids directly at the pituitary level, in the control of gonadotropins secretion. Anandamide is also CB2R agonist, but these receptors are mainly found on immune cells, so in this study we did not analyze *CNR2* gene expression in AP explants. There is evidence that AEA can also operate through other receptors or t ion channels, such as transient receptor potential channels of vanilloid type 1 (TRPV1) and the orphan receptor, GPR55 [46].

## 5. Conclusions

Summarizing, in our ex vivo experiment it was indicated that endogenous cannabinoid AEA in a concentration of 100 µM inhibited GnRH stimulated LH and FSH secretion from the AP explants from anestrous ewes. It was found that AEA influences both LH and FSH gene expressions as well as gonadotropins release. AEA also affected GnRHR synthesis and expression. The presence of CB1R transcript in the AP explants was also demonstrated. However, we did not find any significant influence of AEA on the CNR1 gene expression. Taking into account that AEA undergoes hydrolysis by FAHH to arachidonic acid, it could not be excluded that the effect of AEA on investigated hormones release is mediated by arachidonic acid. This problem requires further, more detailed, research. Therefore, it can be suggested that some negative effects of cannabinoids on the HPG axis activity could be a result of direct action at the pituitary level. It should be also underlined that in the present study, we have used ovine AP explants, as sheep is a popular animal model in biomedical research, as well as in studies on the mechanism influencing gonadotropins secretion [47].

## Figures and Tables

**Figure 1 animals-10-00706-f001:**
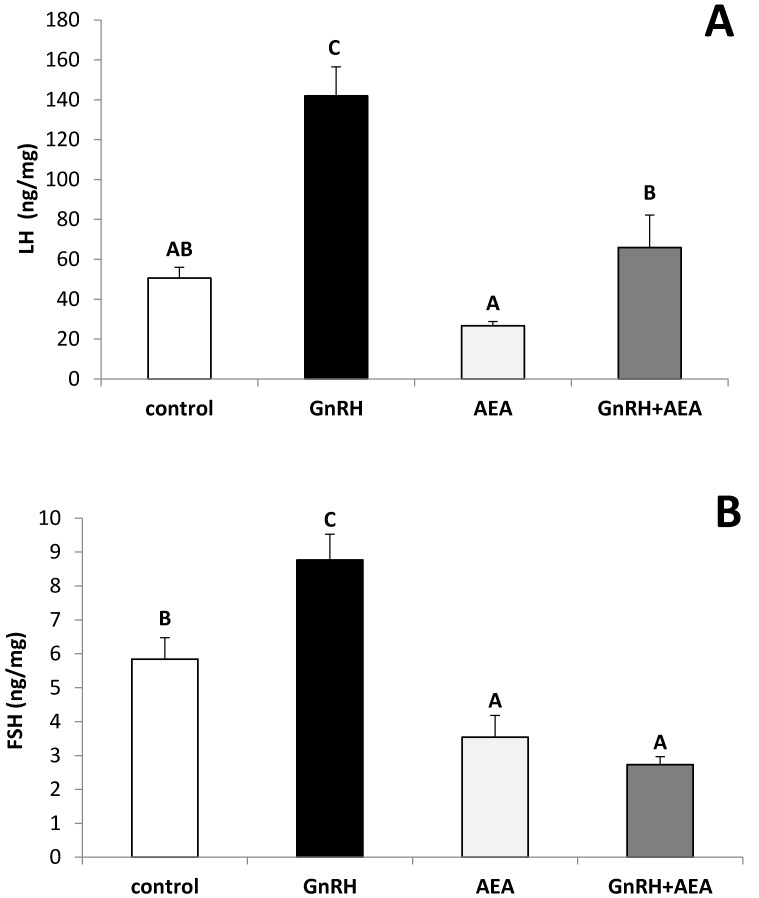
The effects of GnRH (100 pmol/mL) and anandamide (AEA) (100 µM) on luteinizing hormone (LH) (panel **A**) and follicle-stimulating hormone (FSH) (panel **B**) release from the anterior pituitary (AP) explants (*n* = 6 per group) from anestrous ewe. All data are presented as the mean (± S.E.M.). ABC—bars with different superscripts are significantly different from each other accordingly to two-way ANOVA with post-hoc Fisher test at P at least *p* ≤ 0.05.

**Figure 2 animals-10-00706-f002:**
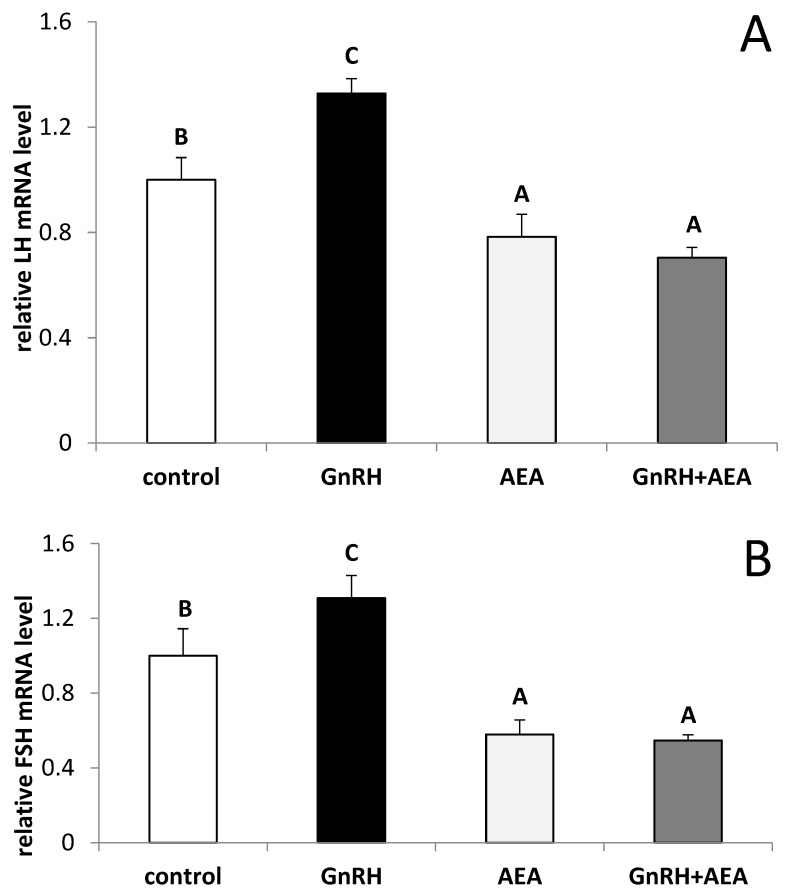
The effects of GnRH (100 pmol/mL) and AEA (100 µM) on the gene expression of β subunits of luteinizing hormone (LH) (panel **A**) and follicle-stimulating hormone (FSH) (panel **B**) in the AP explants (*n* = 6 per group) from anestrous ewes. All data are presented as the mean (± S.E.M.). ABC—bars with different superscripts are significantly different from each other accordingly to two-way ANOVA with post-hoc Fisher test at P at least *p* ≤ 0.05.

**Figure 3 animals-10-00706-f003:**
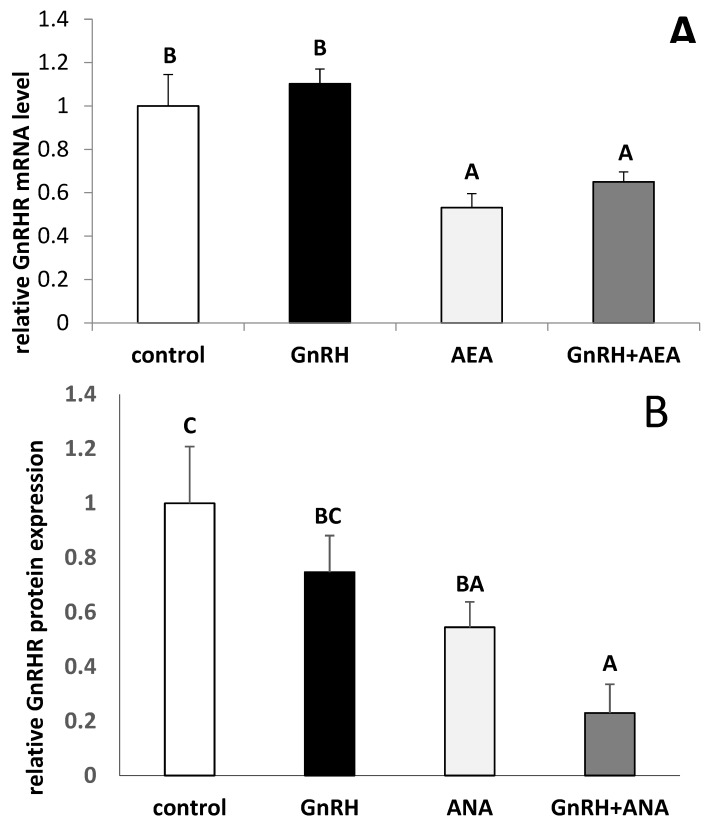
The effects of GnRH (100 pmol/mL) and AEA (100 µM) on the gene (panel **A**) and protein (panel **B**) expression of GnRH receptor (GnRHR) in the AP explants (*n* = 6 per group) from anestrous ewes. All data are presented as the mean (± S.E.M.). ABC—bars with different superscripts are significantly different from each other accordingly to two-way ANOVA with post-hoc Fisher test at *p* ≤ 0.05.

**Figure 4 animals-10-00706-f004:**
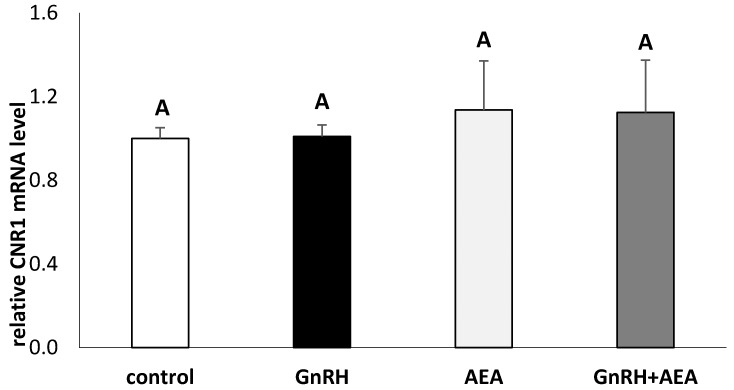
The effects of GnRH (100 pmol/mL) and AEA (100 µM) on the gene expression of cannabinoid type I receptor (CNR1) in the AP explants (*n* = 6 per group) from anestrous ewes. All data are presented as the mean (± S.E.M.). ABC—bars with different superscripts are significantly different from each other accordingly to two-way ANOVA with post-hoc Fisher test at *p* ≤ 0.05.

**Table 1 animals-10-00706-t001:** List of full names and abbreviations of all genes analyzed by Real-Time PCR.

GenBank Acc. No.	Gene	Amplicon Size [bp]	Forward/ Reverse	Sequence 5′→3′	Reference
NM_001034034	*GAPDH * glyceraldehyde-3-phosphate dehydrogenase	134	forward	AGAAGGCTGGGGCTCACT	[29]
			reverse	GGCATTGCTGACAATCTTGA	
U39357	*ACTB * beta actin	168	forward	CTTCCTTCCTGGGCATGG	[29]
			reverse	GGGCAGTGATCTCTTTCTGC	
BC108088.1	*HDAC1 * histone deacetylase1	115	forward	CTGGGGACCTACGGGATATT	[29]
			reverse	GACATGACCGGCTTGAAAAT	
NM_001009397	*GnRHR * gonadotropin-releasing hormone receptor	150	forward	TCTTTGCTGGACCACAGTTAT	[30]
			reverse	GGCAGCTGAAGGTGAAAAAG	
U02517	***GnRH*** gonadotropin-releasing hormone	123	forward	GCCCTGGAGGAAAGAGAAAT	[30]
			reverse	GAGGAGAATGGGACTGGTGA	
X52488	*LHB* luteinizing hormone beta-subunit	184	forward	AGATGCTCCAGGGACTGCT	[30]
			reverse	TGCTTCATGCTGAGGCAGTA	
X15493	*FSHB* follicle stimulating hormone beta-subunit	131	forward	TATTGCTACACCCGGGACTT	[30]
			reverse	TACAGGGAGTCTGCATGGTG	
NM_001242341	*CNR1* cannabinoid receptor 1	104	forward	GAGGACCGGGGGATGC	Originally designed
			reverse	CCGTCGAGGATGGACTTC	

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
