# Peer review of "The Influence of Anandamide on the Anterior Pituitary Hormone Secretion in Ewes—Ex Vivo Study"

_animals, 2020, doi:10.3390/ani10040706_

Round 1

Reviewer 1 Report

I have minor corrections below;

Figures- provide n value for the sample size per bar graph.

Could there be an influence on the relative size difference of the explants on the hormone and gene expression levels?  

Author Response

Dear Sir,

We greatly appreciate all given comments and suggestions. All changes in the manuscript text are made using red type. Our detailed reply for Your comments is below.

Point 1: Figures - provide n value for the sample size per bar graph

I insert in all figures descriptions the value n=6 per group (line 205, 218, 229 and 239)

Point 2: Could there be an influence on the relative size diference of the explants on the hormone and gene expression levels?

In the experiment each AP was dissected and divided into four fragments (explants). We always try to keep all explants fairly even, weight - balanced. But oboviously it isn't possible to have all explants exactly in the same size. Making calculation of hormones release it has always been converted to mg of tissue. Gene expression was calculated always as a relative value, and the results are presented in arbitrary units, as the ratio of the target gene expression to the mean expression of the housekeeping (always the same) genes. However analyzing results we did not observed that there is influence of size of explant on the hormone release and gene expression.

Reviewer 2 Report

Corrections of English (grammar, word-spacing, etc.) should be modified on lines: 10, 11, 13, 15, 21, 28, 35, 36, 48, 49, 53, 67, 68, 70, 72, 86, 99, 111, 118, 127, 140, 160, 170, 209, 210, 224, 231, 240, 242, 250, 251, 258, 260, 268, 273, 275, 285, 286, 297, 310, 312, 315, 319, 371, 409, 411, 412.

The indentation in the paragraphs should be revised.

Simple Summary: The scientific name of a plant "Cannabis sativa" should be properly written.

Introduction: At the end of the introduction (lines 72 to 75) please clarify. Objectives are not clear redacted and the last sentence seems to be unfinished.

Materials and Methods; Radioimmunoassay for LH: If the work is done with ex-situ explants how do they measure the plasma concentration of LH? In the same way, the concentration of FSH was determined in plasma or in the medium? Please clarify.

Statistical analysis and Results:

In Statistical analysis (line 190), statistical significance was defined as P < 0.05. However, it seems that different values of P have defined. In the results, the related figure (Fig1), the value is P≤ 0.05. However, in line 194, the significance for GnRH is expressed as P≤ 0.001. Please clarify these data.

The same situation can be seen in point 3.2 (The ex-vivo influence of GnRH and anandamide on the LHβ and FSHβ gene expression in the AP explants, Fig2), where GnRH has P≤ 0.01.

Captions figures: please consider to change the term "Fig XB" to "Fig.X panel B" in order to clarify the caption of each figure

Author Response

Dear Sir,

I attach my response as a pdf file.

Sincerely D. Tomaszewska-Zaremba

Round 2

Reviewer 2 Report

Please verify spelling in references lines 370 (Herkenham, M,; Lynn), 411 (A.; Billi, S.; Franchi, A.; McCann, S.M.; Rettor,i V. ) and 419 (Suppression ofpulsatile LH).